# Scaling Prompt Synthesis for Large Language Model Reasoning

Xueliang Zhao [1]   Wei Wu [2 3]   Jian Guan [4]   Zhuocheng Gong [3]   Lingpeng Kong [1]

## Abstract

Large language models (LLMs) are evolving from conversational systems into strong reasoners for tasks such as Olympiad mathematics and competitive programming. While scaling parameters and test-time computation has driven progress, a key bottleneck is the lack of high-quality training problems: human-curated datasets are costly and limited, while existing synthetic corpora are often too easy or narrow. PromptCoT showed that injecting rationales into prompt synthesis increases problem difficulty. Building on this, we present PROMPTSCALE, a scalable framework that replaces hand-crafted heuristics with an expectation-maximization (EM) loop, where rationales are iteratively refined to guide prompt construction. This produces problems that are both harder and more diverse than prior corpora. The synthetic prompts support two post-training regimes: (1) *Self-Play*, where strong models improve autonomously via verifiable feedback without stronger teachers; and (2) *Supervised Fine-Tuning (SFT)*, where weaker models learn from teacher-distilled traces. Extensive experiments demonstrate the effectiveness of this approach. In self-play, applying PROMPTSCALE to `Qwen3-30B-A3B-Thinking-2507` sets new state-of-the-art results *at the 30B scale*, with +4.4, +4.8, and +5.3 on AIME 24/25 and HMMT 25, +6.1 and +5.0 on LiveCodeBench v5/v6, and +35 Elo on Codeforces. In SFT, training `Qwen2.5-7B-Instruct` solely on synthetic prompts boosts accuracy to 73.1 (AIME 24), 65.6 (AIME 25), and 53.4 (LiveCodeBench v5), surpassing models trained on human or hybrid data. Analyses further confirm that PROMPTSCALE yields fundamentally harder and distributionally

distinct problems. These results establish prompt synthesis as a new axis for scaling reasoning and position PROMPTSCALE as a scalable foundation for future open-source models. The implementation is available at https://github.com/inclusionAI/PromptCoT.

## 1. Introduction

Large language models (LLMs) have advanced from basic chatbots (Ouyang et al., 2022; Achiam et al., 2023; Hurst et al., 2024) to sophisticated reasoners (Jaech et al., 2024; Guo et al., 2025) capable of addressing complex tasks (e.g., Olympiad-level mathematics) through structured problem decomposition, deliberate planning, and reflective self-correction—all without reliance on external knowledge. More recently, the LLM community has begun to explore the development of agentic intelligence (Moonshot, 2025; Zeng et al., 2025), which aims to endow LLMs with extended capacities to interface with external tools and environments, thereby enabling the autonomous pursuit and completion of more demanding tasks (Phan et al., 2025). Along this trajectory, while large-scale pre-training remains the primary driver of model capability (Gandhi et al.), scaling test-time computation is emerging as a complementary avenue for further enhancing LLM intelligence (Snell et al., 2024).

As test-time scaling continues to attract growing attention within the LLM community, we posit that two lines of research are poised to become the primary technical drivers of this emerging paradigm. The first is *Reinforcement Learning* (RL) (Guo et al., 2025; Schulman et al., 2017), which offers a principled framework for shaping LLM output distributions via carefully designed reward signals, thereby enabling models to learn effectively from experience (Silver & Sutton, 2025). The second, which has received comparatively less attention but may prove even more consequential, is *Task Synthesis* (Yang et al., 2025b; Li et al., 2025; Zhao et al., 2025a; Liu et al., 2025)—automatic and scalable methods for generating task data that constitute the training ground for RL and thus initiate the process of learning from experience.

While human-crafted data continues to play an important role in LLM training (Moshkov et al., 2025; Ahmad et al.,

---

[1]The University of Hong Kong [2]Ant International [3]Ant Group [4]Alibaba Group. Correspondence to: Wei Wu <wuwei19850318@gmail.com>, Lingpeng Kong <lpk@cs.hku.hk>.

*Proceedings of the 43rd International Conference on Machine Learning*, Seoul, South Korea. PMLR 306, 2026. Copyright 2026 by the author(s).

2025), we argue that synthesized task data will become increasingly central to advancing the frontier of LLM intelligence. The shift is driven by two factors: (1) as LLMs grow more capable, the quality of synthesized data will correspondingly improve; and (2) the tasks posed to LLMs will grow substantially more challenging, with their scope expanding dramatically (e.g., in the pursuit of agentic intelligence). Consequently, reliance on human-generated and curated task data will become prohibitively costly, making scalable synthesis indispensable.

In a broad sense, task synthesis may encompass multiple dimensions, including task definition, problem generation, solution synthesis, environment construction, and even the development of evaluation protocols. In this work, we focus on *Prompt Synthesis* as a simplified yet representative form of task synthesis, where an instance of a task is expressed as a textual prompt[1]. Although prompt synthesis has contributed to recent advances in LLMs (Yang et al., 2025a; Zeng et al., 2025; Moonshot, 2025; Adler et al., 2024), the underlying technical details are often scarcely disclosed. Publicly available studies, by contrast, typically rely on deliberately engineered prompts to guide powerful LLMs in generating new prompts (Wang et al., 2023; Xu et al., 2024; Huang et al., 2025b; Tang et al., 2024; Yue et al., 2024; Li et al., 2024; Luo et al., 2024; Wei et al., 2024; Huang et al., 2025a; Xu et al., 2025; Liu et al., 2025; Yang et al., 2025b) -an approach that is neither learnable or scalable. As a result, the datasets produced often fall short in difficulty (Zhao et al., 2025a) and diversity, particularly with respect to expression variation (Xu et al., 2024) and domain coverage (i.e., being restricted to one specific domain). Recently, Zhao et al. (2025a) introduces "rationale"-a form of "thinking process"-into prompt synthesis, demonstrating substantial improvements in the difficulty of synthesized prompts. This method, however, remains dependent on human-engineered prompts and is confined to the mathematical domain. Consequently, we believe that a fully open-sourced pipeline that integrates advanced prompt synthesis and post-training methods would provide a useful resource for both advancing established capabilities (e.g., mathematical reasoning) and supporting exploration of emerging areas in LLM reasoning (e.g., agentic intelligence, scientific discovery, and beyond).

In this work, we present PROMPTSCALE, a principled and scalable framework for prompt synthesis that extends the concept–rationale–prompt paradigm beyond the constraints of PromptCoT (Zhao et al., 2025a). Our method introduces an EM-based optimization procedure in which rationales are iteratively refined to guide prompt construction, thereby generating more challenging and diverse problems across both mathematics and programming domains. Unlike prior approaches that rely heavily on hand-crafted instructions or domain-specific heuristics, PROMPTSCALE is fully learnable and domain-agnostic, enabling synthesis pipelines that can generalize to new reasoning tasks with minimal human intervention.

We demonstrate the effectiveness of PROMPTSCALE through comprehensive experiments under two post-training regimes: (1) Self-Play, where synthesized problems provide automatically verifiable feedback that enables models to improve without stronger external teachers; and (2) Supervised Fine-Tuning (SFT), where weaker base models are trained from complete reasoning traces distilled from a teacher. In the self-play setting, PROMPTSCALE delivers substantial improvements over strong Qwen3 baselines and outperforms all existing open-source corpora. For example, when applied to `Qwen3-30B-A3B-Thinking-2507`, our method improves accuracy on AIME 24 from 87.7 to **92.1**, on AIME 25 from 85.0 to **89.8**, and on HMMT Feb 25 from 71.4 to **76.7**, while also achieving notable gains on LiveCodeBench v5 (+6.1) and v6 (+5.0). Overall, these results establish new state-of-the-art performance *at the 30B parameter scale* across all six benchmarks. In the SFT setting, we initialize from `Qwen2.5-7B-Instruct` and train exclusively on prompts synthesized by PROMPTSCALE. Despite relying solely on synthetic problems, the resulting model achieves consistent improvements over both human-curated and hybrid datasets. For instance, accuracy improves from 12.8 to **73.1** on AIME 24, from 8.0 to **65.6** on AIME 25, from 2.7 to **46.5** on HMMT Feb 25, from 14.7 to **53.4** on LiveCodeBench v5, and from 13.7 to **48.9** on LiveCodeBench v6, while also achieving the strongest Codeforces performance (706 → **1815**). Extended investigation further demonstrates that PROMPTSCALE produces prompts that exhibit striking semantic differences from human-created ones and are considerably more challenging than those generated by PromptCoT.

Our contributions can be summarized as follows:

- We introduce a new rationale-driven prompt synthesis method, PROMPTSCALE, which integrates EM optimization to jointly refine rationale generation and prompt construction. This framework moves beyond handcrafted instructions and scales naturally to multiple domains.

- We show how synthesized prompts can be effectively used for LLM post-training. Specifically, we present two complementary regimes—self-play and supervised fine-tuning—and demonstrate consistent improvements over strong Qwen3 and Qwen2.5 baselines across six challenging benchmarks.

- We contribute a large-scale corpus of synthetic prompts

---

[1]A prompt may also incorporate multimodal elements such as images, videos, or tools. We defer the exploration of multimodal prompt synthesis to future work.

fundamentally different from existing open-source datasets, with both distributional and difficulty analyses confirming that PROMPTSCALE produces problems that are more diverse and substantially more challenging. This resource provides a new foundation for advancing reasoning in open-source LLMs.

## 2. Preliminaries

### 2.1. Problem Formulation: Prompt Synthesis

We consider the setting where a prompt is constructed from a set of underlying concepts. Let $\mathcal{C}$ denote the concept space and $\mathcal{X}$ the prompt space. Each prompt $x \in \mathcal{X}$ is associated with a subset of concepts $\mathbf{c} \subseteq \mathcal{C}$. Prompt synthesis is then formulated as modeling the conditional distribution $p(x \mid \mathbf{c})$, with the objective of generating a prompt instance $x$ that faithfully reflects the specified concepts and ultimately enhances the reasoning capabilities of LLMs.

### 2.2. The PromptCoT Framework

A central challenge in prompt synthesis is that mapping concepts directly to prompts often yields underspecified or brittle problem statements (Zhao et al., 2025a). To address this, PromptCoT (Zhao et al., 2025a) introduces an intermediate latent variable—the rationale. Instead of modeling $p(x \mid \mathbf{c})$ directly, PromptCoT factorizes it via rationales $z$, yielding:

$$p(x \mid \mathbf{c}) = \sum_z p(x \mid z, \mathbf{c})\, p(z \mid \mathbf{c}). \qquad (1)$$

The novelty of PromptCoT lies in its reframing of prompt synthesis: instead of constructing prompts directly from concepts, the process is mediated through rationales, such that prompts are derived jointly from the concepts and their explanatory structures. Rationales act as "thinking process" that grounds the concepts, decomposes them into intermediate explanatory steps, and guides the construction of more robust prompts. This shift has proven particularly effective in the mathematical domain (Zhao et al., 2025a), where incorporating rationales substantially increases the difficulty of synthesized prompts and, in turn, yields significant improvements on challenging mathematical reasoning tasks.

While PromptCoT pioneers the exploration of the "thinking mode" in prompt synthesis, its reliance on human-crafted instructions and validation within a single domain highlight important limitations. The promising yet constrained empirical results, together with the technical limitations, motivate further investigation toward a more principled approach capable of enhancing the reasoning abilities of LLMs across broader tasks.

## 3. PromptScale

An overview of the method is provided in Figure 1, where a rationale generation model $q_\phi(z \mid \mathbf{c}, x)$ and a prompt generation model $p_\theta(z, x \mid \mathbf{c})$ are jointly trained under an Expectation–Maximization (EM) procedure.

In the E-step, $q_\phi(z \mid \mathbf{c}, x)$ is updated to assign higher probability to rationales that better connect concepts and prompts; while in the M-step, $p_\theta(z, x \mid \mathbf{c})$ is trained on concept–rationale–prompt triples $(\mathbf{c}, z, x)$, where rationales are inferred by the latest $q_\phi(z \mid \mathbf{c}, x)$. The objective is to maximize $\mathbb{E}_{q_\phi(z \mid \mathbf{c}, x)}[\log p_\theta(z, x \mid \mathbf{c})]$ over observed $(\mathbf{c}, x)$. Alternating these two steps progressively improves both $q_\phi(z \mid \mathbf{c}, x)$ and $p_\theta(z, x \mid \mathbf{c})$, creating an iterative loop in which rationales guide prompt construction and prompt synthesis, in turn, informs the discovery of more effective rationales.

### 3.1. Variational Perspective

Direct optimization of the marginal likelihood (Eq. 1) is infeasible due to the summation over rationales. To address this, we introduce the approximate posterior $q_\phi(z \mid \mathbf{c}, x)$, yielding the evidence lower bound (ELBO):

$$\begin{aligned} \log p_\theta(x \mid \mathbf{c}) \;\geq\; & \mathbb{E}_{q_\phi(z \mid \mathbf{c}, x)}\left[\log p_\theta(z, x \mid \mathbf{c})\right] \\ & - \; \mathrm{KL}\left(q_\phi(z \mid \mathbf{c}, x) \| p_\theta(z \mid \mathbf{c})\right). \end{aligned} \qquad (2)$$

This decomposition aligns naturally with the EM interpretation. In the E-step, maximizing the ELBO with respect to $q_\phi$ is equivalent to minimizing its KL divergence to the true posterior, whose optimal form is

$$q_\phi^\star(z \mid \mathbf{c}, x) \;\propto\; p_\theta(z, x \mid \mathbf{c}) \;=\; p_\theta(x \mid z, \mathbf{c})\, p_\theta(z \mid \mathbf{c}). \qquad (3)$$

Thus, rationales are rewarded in proportion to their joint likelihood: they must align with the concepts while also being predictive of valid prompts.

In the M-step, with $q_\phi$ fixed, maximizing the ELBO reduces to maximizing the expected joint log-likelihood under the current posterior:

$$\mathbb{E}_{q_\phi(z \mid \mathbf{c}, x)}\left[\log p_\theta(z, x \mid \mathbf{c})\right]. \qquad (4)$$

This expectation provides the training signal for the prompt generation model: given observed pairs $(\mathbf{c}, x)$, rationales are sampled from $q_\phi(z \mid \mathbf{c}, x)$, and the model is updated to maximize the likelihood of reproducing the corresponding rationale–prompt pairs.

These updates establish a principled optimization loop: rationale generation is refined by rewards derived from prompt generation, and prompt generation is updated using rationales proposed by the posterior. The tight correspondence between the ELBO, the reward design, and the EM procedure ensures that both components improve in tandem, with

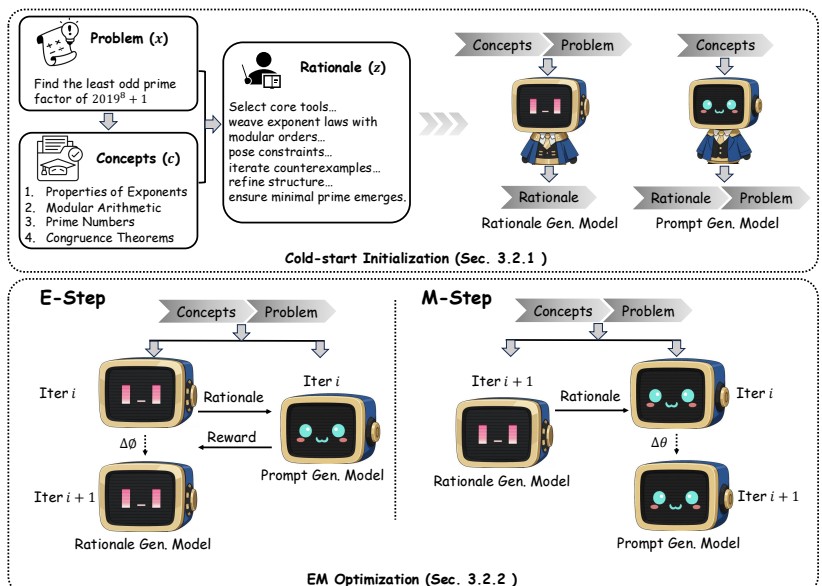

*Figure 1.* Overview of PROMPTSCALE. We begin with open-source problems and annotate their associated concepts and rationales, forming concept–rationale–problem triples that provide cold-start data for the rationale generation model and the prompt generation model. During EM optimization, the **E-step** updates the rationale generation model by maximizing the reward defined in Eq. 5, while the **M-step** updates the prompt generation model to better align with the generated rationales.

rationales functioning as latent bridges between concepts and prompts.

### 3.2. Practical Implementation

PROMPTSCALE is implemented as a two-phase training procedure, consisting of a cold-start initialization phase (§3.2.1) followed by an iterative EM optimization phase (§3.2.2).

#### 3.2.1. COLD-START INITIALIZATION

To bootstrap both the rationale generation model $q_\phi$ and the prompt generation model $p_\theta$, we follow the data construction procedure introduced in PromptCoT (Zhao et al., 2025a). Specifically, we begin from open-source problem sets in mathematics and programming, and construct associated concept and rationale annotations. For each concept–problem pair $(\mathbf{c}, x)$, annotations are generated by prompting four high-capacity instruction-tuned models: `Llama-3.1-70B-Instruct`, `phi-4`, `Qwen2.5-32B-Instruct` and `Qwen2.5-72B-Instruct`. This procedure yields a seed corpus of concept–rationale–problem triples, which is then used to warm-start $q_\phi$ and $p_\theta$ via maximum likelihood estimation.

#### 3.2.2. EM OPTIMIZATION

With the initial models in place, we proceed to optimize them jointly through the EM loop. In the E-step, the rationale generation model $q_\phi(z \mid \mathbf{c}, x)$ is updated via reinforce-

ment learning, with the reward defined as

$$R(\mathbf{c}, x, z) = \log p_\theta(x \mid z, \mathbf{c}) + \log p_\theta(z \mid \mathbf{c}), \quad (5)$$

which corresponds to the joint log-likelihood $\log p_\theta(z, x \mid \mathbf{c})$. Given a concept–prompt pair $(\mathbf{c}, x)$, rationales are sampled from $q_\phi(z \mid \mathbf{c}, x)$. For each pair, we draw eight candidate rationales and select the one with the highest reward; the parameters $\phi$ are then updated via supervised fine-tuning on this chosen rationale, ensuring that the model learns to prefer rationales that are both faithful to the concepts and predictive of valid prompts. In the M-step, the prompt generation model $p_\theta(z, x \mid \mathbf{c})$ is trained using the objective described in Eq. 4, leveraging rationales sampled from the current posterior. The two models are alternately updated until convergence, ensuring that $q_\phi$ aligns with the posterior distribution over rationales while $p_\theta$ adapts to rationales discovered in each iteration.

## 4. LLM Post-Training with Synthesized Prompts

Armed with $p_\theta(z, x \mid \mathbf{c})$, we can construct a synthesized prompt dataset and post-train an LLM to enhance its reasoning capabilities. In PromptCoT, post-training is carried out through SFT, where reasoning traces and answers are distilled from strong teacher models. PROMPTSCALE, by contrast, underscores the inherent limitations of SFT: as reasoning capabilities continue to advance, state-of-the-art LLMs demand ever-stronger teachers—models that are often inaccessible for distillation (e.g., proprietary models) or prohibitively expensive to employ (e.g., due to their massive pa-

rameter scale). To overcome these barriers, PROMPTSCALE moves beyond SFT and establishes a more broadly applicable post-training strategy, with a self-play regime at its core—enabling models to improve autonomously without dependence on increasingly powerful external teachers.

**Self-Play.** For models that already possess strong reasoning capabilities (e.g., `Qwen3-30B-A3B-Thinking-2507`), the primary bottleneck shifts from the quality of supervision to the absence of even stronger teacher models for distillation. PROMPTSCALE overcomes this limitation by employing a self-play regime, where learning is guided by automatically verifiable feedback rather than relying on external traces.

Formally, given a collection of prompts $\{x_i\}_{i=1}^N$ with automatically verifiable signals, [2] a base LLM $\mathcal{M}_\psi$ generates candidate solutions:

$$y_i^{(j)} \sim \mathcal{M}_\psi(\cdot \mid x_i), \quad j = 1, \ldots, k.$$

Each candidate is then assigned a scalar reward $u(x_i, y_i^{(j)}) \in \mathbb{R}$ derived from the corresponding verification signal. The general self-play objective is to update parameters $\psi$ to maximize expected reward:

$$\max_\psi \ \frac{1}{N} \sum_{i=1}^N \mathbb{E}_{y \sim \mathcal{M}_\psi(\cdot \mid x_i)} \big[ u(x_i, y) \big].$$

This framework is compatible with multiple optimization strategies, including PPO (Schulman et al., 2017), GRPO (Shao et al., 2024), and DPO (Rafailov et al., 2023). In all cases, the core mechanism is the same: the model improves by iteratively generating solutions, receiving feedback, and updating itself, without dependence on external teachers or human annotations.

**Supervised Fine-Tuning.** For models that are not yet proficient in complex reasoning (e.g., `Qwen2.5-7B-Instruct`), self-play is no longer applicable. Instead, we employ a strong teacher model to generate complete reasoning traces for the synthesized prompts. The student model is then trained in a supervised manner on these problem–trace pairs, thereby acquiring reasoning behaviors directly from teacher demonstrations. This ensures that even weaker models can benefit effectively from PROMPTSCALE through explicit trajectory-level supervision.

---

[2]This process requires no additional human annotation as detailed in §B.4.

## 5. Experiments

### 5.1. Benchmarks

We evaluate our models across six diverse benchmarks spanning mathematics and code generation. Each dataset targets a different aspect of reasoning or problem-solving: (1) **AIME 24** is an annual high-school mathematics competition from the American Mathematics Competitions (AMC) series. The 2024 set contains 30 problems designed to test advanced mathematical reasoning, covering algebra, number theory, geometry, and combinatorics; (2) **AIME 25** is the 2025 edition of the AIME competition, with a new set of 30 problems of comparable style and difficulty; (3) **HMMT Feb 25** is a premier international high-school mathematics tournament. We evaluate on the February 2025 contest set, which contains 30 problems spanning diverse mathematical domains; (4) **LiveCodeBench v5 (2408–2502)** (Jain et al., 2025) is a live-updated benchmark that sources real-world coding problems from LeetCode, AtCoder, and CodeForces. We use the problems released between August 2024 and February 2025, totaling 279 problems; (5) **LiveCodeBench v6 (2502–2505)** (Jain et al., 2025) extends the dataset with newer problems released between February 2025 and May 2025, containing 131 problems. This split highlights temporal generalization by evaluating on problems unseen in v5; and (6) **Codeforces** (Quan et al., 2025) consists of 408 real-world competitive programming problems drawn from the Codeforces platform. Problems cover a wide spectrum of algorithmic reasoning challenges and reflect authentic conditions faced in live contests.

### 5.2. Evaluation Metrics

We adopt **pass@1 accuracy** as the primary evaluation metric across all benchmarks. For mathematical benchmarks (AIME 24, AIME 25, HMMT Feb 25), we report **avg@16**, where pass@1 accuracy is averaged over 16 independent generations per problem to reduce evaluation variance. For programming benchmarks (LiveCodeBench v5, LiveCodeBench v6), we report the standard **pass@1**, computed from a single generation per problem. In both cases, correctness is determined strictly: a mathematical output must exactly match the ground-truth final boxed answer, while a programming output must pass all provided functional unit tests.

For the Codeforces benchmark, we follow the evaluation protocol of Luo et al. (2025a) and compute Elo ratings to evaluate the performance gap between models and competitive programming experts. Each problem is attempted with up to 8 independent generations, and the Elo system aggregates outcomes to produce a competitive programming–style performance measure.

*Table 1.* Evaluation results on six benchmarks under the Self-Play setting using models with **4B** parameters. Bold values denote the best performance for each benchmark, while underlined values denote the second-best.

| Model | AIME 24 | AIME 25 | HMMT Feb 25 | LiveCodeBench v5 (2408–2502) | LiveCodeBench v6 (2502–2505) | Codeforces |
|---|---|---|---|---|---|---|
| Qwen3-4B-Thinking-2507 | 85.2 | 81.3 | 55.5 | 63.8 | 55.2 | 1852 |
| OpenCodeReasoning | 83.1 | 78.5 | 50.4 | 64.4 | 57.1 | 1867 |
| OpenMathReasoning | 85.3 | 83.0 | 56.8 | 59.7 | 48.5 | 1826 |
| OpenThoughts3 | 84.7 | 80.6 | 54.2 | 65.2 | 54.4 | 1846 |
| OpenR1 | 84.6 | 80.9 | 56.7 | 63.0 | 54.6 | 1829 |
| PromptCoT | 85.3 | 81.8 | 58.6 | 64.5 | 56.7 | 1878 |
| PROMPTSCALE | **87.3** | **85.0** | **66.5** | **67.7** | **61.1** | **1934** |

*Table 2.* Evaluation results on six benchmarks under the Self-Play setting using models with **30B-A3B** parameters. Bold values denote the best performance for each benchmark, while underlined values denote the second-best.

| Model | AIME 24 | AIME 25 | HMMT Feb 25 | LiveCodeBench v5 (2408–2502) | LiveCodeBench v6 (2502–2505) | Codeforces |
|---|---|---|---|---|---|---|
| Qwen3-30B-A3B-Thinking-2507 | 87.7 | 85.0 | 71.4 | 68.1 | 66.0 | 2044 |
| OpenCodeReasoning | 85.0 | 81.1 | 64.9 | 70.8 | 67.4 | 2048 |
| OpenMathReasoning | 87.9 | 86.1 | 72.2 | 65.7 | 58.4 | 2022 |
| OpenThoughts3 | 85.7 | 84.7 | 70.0 | 68.3 | 67.2 | 2019 |
| OpenR1 | 86.3 | 84.9 | 68.9 | 68.3 | 63.7 | 2045 |
| PromptCoT | 86.8 | 87.4 | 72.0 | 69.7 | 67.8 | 2051 |
| PROMPTSCALE | **92.1** | **89.8** | **76.7** | **74.2** | **71.0** | **2079** |

## 5.3. Baselines

We evaluate against two complementary experimental settings: *self-play* and *SFT*. Both settings rely on the same family of problem curation baselines, but differ in base models and in how the released corpora are utilized.

**Self-Play.** We compared PROMPTSCALE against a set of open-source corpora comprising large-scale mathematical and programming problems, as well as mixed collections. Prompts from each corpus were used to train `Qwen3-4B-Thinking-2507` and `Qwen3-30B-A3B-Thinking-2507` (Yang et al., 2025a) under the self-play setting. Specifically, we considered the following corpora: (1) **OpenCodeReasoning** (Ahmad et al., 2025), containing curated competitive programming problems from Codeforces, AtCoder, and LeetCode; (2) **OpenMathReasoning** (Moshkov et al., 2025), consisting of competition-style mathematics problems from AoPS and related repositories; (3) **OpenThoughts3** (Guha et al., 2025), a collection of curated cross-domain questions augmented with synthetic generation; (4) **OpenR1** (Hugging Face, 2025), comprising curated problems from Codeforces and NuminaMath variants; and (5) **PromptCoT** (Zhao et al., 2025a;b), a corpus of mathematics and programming problems augmented through the concept–rationale–prompt paradigm. Since the original corpora were designed for large-scale SFT and were therefore too large for self-play, we downsampled their prompt sets to ensure all methods operated on comparable data sizes.

**SFT.** The base model used is `Qwen2.5-7B-Instruct` (Yang et al., 2024). For PromptCoT, we adhered to the training protocol outlined

in Zhao et al. (2025b). For the other baselines, we directly evaluated the officially released checkpoints without any additional training or modifications. In addition, we include an additional experiment in Appendix C.5, where all baseline teacher models are standardized to `GPT-OSS-120B (medium)`, enabling a controlled comparison with our method.

## 5.4. Main Results

**Self-Play Setting.** Table 1 and Table 2 report the results of self-play experiments, from which several noteworthy observations emerge: (1) Incorporating rationales proves beneficial for synthesizing high-quality prompts, even when implemented in a straightforward manner through prompt engineering (i.e., PromptCoT). (2) PROMPTSCALE exhibits superior data efficiency: in the 4B setting, it attains superior results using only about 90% of the math prompts and 10% of the code prompts compared to OpenMathReasoning and OpenCodeReasoning. The 30B setting further reduces the required data while achieving even stronger gains. (3) While both OpenThoughts3 and OpenR1 include a substantial proportion of high-quality, human-crafted mathematics and programming questions, they generally fail to provide further gains for strong models such as `Qwen3-4B-Thinking-2507` and `Qwen3-30B-A3B-Thinking-2507`. A plausible explanation is that such data may have already been incorporated into the training of these models. In contrast, despite the already strong capabilities of the base models, prompts synthesized by PROMPTSCALE deliver consistent and significant improvements across a wide range of mathematical and programming tasks, underscoring the considerable potential of prompt synthesis in advancing the frontier of rea-

*Table 3.* Evaluation results on six benchmarks under the SFT setting using models with **7B** parameters. **H** = Human-written, **S** = Synthetic, **Avg. Resp. Len.** = average response length, and **LCB** = LiveCodeBench. Bold values denote the best performance for each benchmark, while underlined values denote the second-best.

| Model | Prompt Source | Teacher Model | Dataset Size | Avg. Resp. Len. | AIME 24 | AIME 25 | HMMT Feb 25 | LCB v5 (2408–2502) | LCB v6 (2502–2505) | Codeforces |
|---|---|---|---|---|---|---|---|---|---|---|
| **Qwen2.5-7B-Instruct** | - | - | - | - | 12.8 | 8.0 | 2.7 | 14.7 | 13.7 | 706 |
| **OpenCodeReasoning** | H | DeepSeek-R1 | 0.75M | 12878.9 | 11.7 | 7.7 | 6.0 | 50.5 | 42.0 | 1648 |
| **OpenMathReasoning** | H | DeepSeek-R1 | 4.92M | 14407.5 | **73.3** | 58.1 | 42.1 | 9.7 | 10.7 | 676 |
| **OpenThoughts3** | H+S | QwQ-32B | 1.20M | 18740.2 | 69.6 | 59.4 | 38.3 | 47.0 | 36.6 | 1630 |
| **OpenR1** | H | DeepSeek-R1 | 0.35M | 14974.3 | 55.6 | 39.2 | 24.6 | 40.5 | 32.8 | 1548 |
| **PromptCoT** | H+S | QwQ-32B | 1.25M | 16323.5 | 71.0 | 60.2 | 41.6 | 47.8 | 43.6 | 1645 |
| **PROMPTSCALE** | S | GPT-OSS-120B (medium) | 4.77M | 7351.2 | 73.1 | **65.6** | **46.5** | **53.4** | **48.9** | **1815** |

*Table 4.* Ablation results under the SFT setting using models with **7B** parameters. Bold numbers indicate the highest performance.

| Model | AIME 24 | AIME 25 | HMMT Feb 25 | LiveCodeBench v5 (2408–2502) | LiveCodeBench v6 (2502–2505) | Codeforces |
|---|---|---|---|---|---|---|
| **PROMPTSCALE (full)** | **73.1** | **65.6** | **46.5** | **53.4** | **48.9** | **1815** |
| – Code-start | 72.1 | 63.9 | 45.3 | 50.4 | 43.6 | 1677 |
| – EM | 69.8 | 59.0 | 41.3 | 44.9 | 44.2 | 1592 |

soning LLMs. (4) Finally, the advantage of PROMPTSCALE over the baseline methods becomes even more pronounced on challenging tasks such as HMMT, a gain that can be attributed to its EM-based rationale generation, which provides stronger inductive signals for synthesizing sufficiently difficult questions for these benchmarks.

**SFT Setting.** We further evaluate PROMPTSCALE under the SFT setting, with results presented in Table 3. From the results, we draw several key observations: (1) With 100% synthesized prompts, PROMPTSCALE delivers strong performance across all benchmarks. Given that prompt synthesis is far more scalable than manual curation, these results highlight the considerable potential of scaling synthetic data for future advances in LLM reasoning. (2) Beyond its superior performance, the dataset produced by PROMPTSCALE also exhibits substantially shorter reasoning traces, leading to notable computational savings during inference. While this property may largely stem from the teacher model, the released dataset provides significant benefits to the broader community due to its scale, quality, and compact responses.

## 6. Discussions

Beyond the comprehensive evaluation across several benchmarks, we further investigate PROMPTSCALE to gain deeper insights into its underlying mechanisms. Specifically, we investigate the following research questions: (1) **RQ1:** How do the individual components of PROMPTSCALE contribute to its overall performance? (2) **RQ2:** How does PROMPTSCALE scale with increasing data in both self-play and SFT settings? (3) **RQ3:** How does the diversity of problems generated by PROMPTSCALE compare to existing datasets? (4) **RQ4:** How does the difficulty of problems produced by PROMPTSCALE compare to those in existing datasets? and (5) **RQ5:** What is the effect of iterative rationale refinement during the EM optimization process on

problem quality and model performance? We leave the discussions to RQ3, RQ4 and RQ5 to Appendix C.

### 6.1. Ablation Study for RQ1

We conducted an ablation study under the SFT setting using 7B models to evaluate the contribution of each component in PROMPTSCALE. Specifically, we considered two variants: (1) removal of the cold-start stage, denoted as "- Code-start"; and (2) removal of the EM optimization stage, denoted as "- EM". The results are summarized in Table 4.

The full PROMPTSCALE consistently outperforms both ablated variants across all six benchmarks, underscoring the necessity of both stages. Excluding the cold-start stage leads to a moderate but consistent performance decline, showing that high-quality initial annotations from large teacher models are essential for stabilizing the subsequent training loop. In contrast, removing the EM optimization stage causes the largest performance drop, particularly on HMMT and Codeforces, demonstrating that the iterative refinement of rationales and prompts is central to the success of our approach.

### 6.2. Scaling Analysis for RQ2

We next examined the scaling behavior of PROMPTSCALE compared to curation-based baselines under both the self-play and the SFT settings. For self-play experiments, we initialized from `Qwen3-4B-Thinking-2507` and progressively increase the number of training examples. For SFT experiments, we initialized from `Qwen2.5-7B-Instruct`, again varying the amount of training data. To ensure comparability, we randomly sampled each dataset to match the desired scale. We report results on mathematics and programming benchmarks separately, and in all cases we plot the average accuracy across the relevant benchmarks (AIME 24, AIME 25, HMMT Feb 25 for mathematics; LiveCodeBench v5 and v6 for programming). Due to space constraints, the

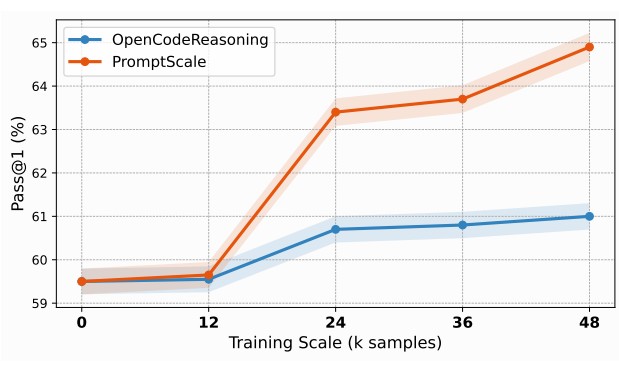

*(a)* Self-play on code benchmarks.

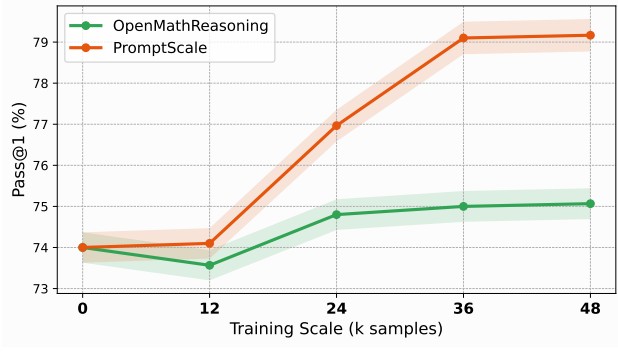

*(b)* Self-play on math benchmarks.

*Figure 2.* Learning curves of PROMPTSCALE compared with baseline methods.

detailed results of the SFT scaling experiments are provided in Appendix C.1.

Figure 2 (a) and Figure 2 (b) illustrate the effect of increasing training examples from 0 to 48k. The curation-based baselines show limited or even unstable improvements, with performance gains largely saturating beyond 24k examples. In contrast, PROMPTSCALE demonstrates consistent scaling: accuracy steadily improves with additional data. This highlights the advantage of EM-guided rationale–prompt synthesis, which produces higher-quality problems that continue to yield benefits at larger scales.

## 7. Related Work

**Prompt Synthesis.** Research on prompt synthesis has evolved alongside the rapid growth of LLMs. Early studies focused on general-purpose methods, such as Wang et al. (2023), who used GPT-3 to generate prompts by imitating a small set of seed instructions, and Xu et al. (2024), who introduced a self-evolution algorithm to increase prompt complexity through ChatGPT. Later work shifted to domain-specific prompt generation to enhance LLM capabilities. For example, Luo et al. adapted the self-evolution framework for mathematical reasoning, while Huang et al. (2025b) and Tang et al. (2024) generated math problems from key concepts. Zhao et al. (2025a) further enhanced this by incorporating a "thinking process" in math problem generation to increase problem difficulty. In programming, Huang et al. (2025a) and Wei et al. (2024) focused on generating coding problems from code snippets. Recently, as interest in agentic intelligence grew, prompt synthesis for task-oriented problems emerged to address data scarcity. For instance, Li et al. (2025) synthesized knowledge-seeking questions from knowledge graphs, Liu et al. (2025) developed an iterative method to increase question difficulty, and Yang et al. (2025b) used a hybrid approach to create code bugs from GitHub repositories. However, a major limitation of prior approaches is their reliance on heuristic or single-pass generation pipelines, which often fail to generate the deep reasoning structures needed for challenging problems. In contrast, our approach uses an EM procedure for rationale–prompt co-generation, where the E-step refines rationales via reward-guided inference and the M-step updates prompts based on these rationales. This iterative process ensures more complex and faithful problem generation in both mathematics and programming.

**Self-Evolution.** The development of self-evolutionary LLMs, where prompt generation and model updating occur in an iterative cycle, has gained attention as an alternative to human-curated data. Notable works include Yuan et al. (2024), who applied the self-instruct framework (Wang et al., 2023) for prompt synthesis and improved the model via supervised fine-tuning, and Zhao et al. (2026), who combined task synthesis with reinforcement learning in an iterative process. PROMPTSCALE shares similarities with these self-evolutionary approaches but differs in that it does not rely on self-judgment or joint optimization of prompt synthesis and model training. While PROMPTSCALE primarily focuses on advancing prompt synthesis, its high-quality prompts and state-of-the-art performance across various tasks provide a foundation for future exploration in self-evolutionary LLMs.

## 8. Conclusions

In this work, we introduced PROMPTSCALE, a principled framework for rationale-driven prompt synthesis that addresses the persistent shortage of high-quality training problems for reasoning-focused LLMs. By formulating concept–rationale–prompt generation as an expectation–maximization procedure, our method iteratively refines rationales and leverages them to construct substantially more difficult and diverse problems across mathematics and programming. Beyond synthesis, we demonstrated how these problems can fuel two complementary post-training regimes: *self-play*, which enables strong mod-

els to improve autonomously through verifiable feedback without reliance on ever-stronger external teachers, and *supervised fine-tuning*, which allows weaker models to acquire reasoning skills from teacher-distilled traces. Extensive experiments show that PROMPTSCALE not only establishes new state-of-the-art results at the 30B scale but also allows 7B models trained purely on synthetic prompts to achieve performance competitive with models trained on human-curated or hybrid datasets. Our analyses further confirm that the synthesized problems differ fundamentally from prior corpora, exhibiting higher difficulty and richer distributional diversity, both of which are essential for advancing LLM reasoning.

## Impact Statement

This paper presents PROMPTSCALE, a framework designed to enhance reasoning capabilities in LLMs by generating diverse and challenging problems. This approach reduces reliance on costly human-curated datasets, enabling significant progress in fields like education, technology, and research. In accordance with established ethical guidelines, this research exclusively utilizes publicly accessible data and information, ensuring that no private or confidential resources are involved.

## Acknowledgements

We extend our gratitude to the HKU NLP group and the anonymous reviewers for their invaluable suggestions, which significantly enhanced this work. This work is partially supported by the joint research scheme of the National Natural Science Foundation of China (NSFC) and the Research Grants Council (RGC) under grant number N_HKU714/21.

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

# A. Proofs of Variational Results

## A.1. Proof of the ELBO (Eq. 2)

We provide a detailed derivation of the evidence lower bound.

*Proof.* Starting from the marginal likelihood (Eq. 1):

$$\log p_\theta(x \mid \mathbf{c}) = \log \sum_z p_\theta(z, x \mid \mathbf{c})$$
$$= \log \sum_z q_\phi(z \mid \mathbf{c}, x) \frac{p_\theta(z, x \mid \mathbf{c})}{q_\phi(z \mid \mathbf{c}, x)}$$
$$= \log \mathbb{E}_{q_\phi(z \mid \mathbf{c}, x)} \left[ \frac{p_\theta(z, x \mid \mathbf{c})}{q_\phi(z \mid \mathbf{c}, x)} \right].$$

Applying Jensen's inequality yields

$$\log p_\theta(x \mid \mathbf{c}) \geq \mathbb{E}_{q_\phi(z \mid \mathbf{c}, x)} \left[ \log \frac{p_\theta(z, x \mid \mathbf{c})}{q_\phi(z \mid \mathbf{c}, x)} \right]$$
$$= \mathbb{E}_{q_\phi(z \mid \mathbf{c}, x)} \left[ \log p_\theta(z, x \mid \mathbf{c}) \right] - \mathbb{E}_{q_\phi(z \mid \mathbf{c}, x)} \left[ \log q_\phi(z \mid \mathbf{c}, x) \right].$$

Adding and subtracting $\mathbb{E}_{q_\phi} \left[ \log p_\theta(z \mid \mathbf{c}) \right]$ gives

$$\log p_\theta(x \mid \mathbf{c}) \geq \mathbb{E}_{q_\phi(z \mid \mathbf{c}, x)} \left[ \log p_\theta(z, x \mid \mathbf{c}) \right] - \mathrm{KL}(q_\phi(z \mid \mathbf{c}, x) \, \| \, p_\theta(z \mid \mathbf{c})) \,,$$

which is precisely Eq. 2. □

## A.2. Proof of the Optimal Posterior (Eq. 3)

We now establish the form of the optimal approximate posterior.

*Proof.* Rewriting Eq. 2, we obtain

$$\log p_\theta(x \mid \mathbf{c}) = \mathcal{L}(q_\phi) + \mathrm{KL}(q_\phi(z \mid \mathbf{c}, x) \, \| \, p_\theta(z \mid \mathbf{c}, x)) \,,$$

where $\mathcal{L}(q_\phi)$ is the ELBO. Since the KL divergence is non-negative, the ELBO is maximized when

$$q_\phi^\star(z \mid \mathbf{c}, x) = p_\theta(z \mid \mathbf{c}, x).$$

By Bayes' rule,

$$p_\theta(z \mid \mathbf{c}, x) = \frac{p_\theta(z, x \mid \mathbf{c})}{p_\theta(x \mid \mathbf{c})} \; \propto \; p_\theta(z, x \mid \mathbf{c}) = p_\theta(x \mid z, \mathbf{c}) \, p_\theta(z \mid \mathbf{c}),$$

which matches Eq. 3. □

# B. Implementation Details

## B.1. Instruction for Concept Extraction

As part of the cold-start stage, this prompt is designed to extract domain-relevant concepts from each seed problem. It instructs the language model to identify the key mathematical or programming concepts that underlie the given task.

---

**Concept Extraction Prompt**

As an expert in educational assessment, analyze this problem:
{problem}
Break down and identify {num_concepts} foundational concepts being tested. List these knowledge points that:

- Are core curriculum concepts typically taught in standard courses,

- Are precise and measurable (not vague like "understanding math"),

- Are essential building blocks needed to solve this problem,

- Represent fundamental principles rather than problem-specific techniques.

Think through your analysis step by step, then format your response as a Python code snippet containing a list of {num_concepts} strings, where each string clearly describes one fundamental knowledge point.

---

### B.2. Instruction for Rationale Generation

In the cold-start stage, the following prompt instructs the language model to produce a detailed reasoning process that serves as the rationale for prompt synthesis.

---

**Rationale Generation Instruction**

Imagine you are an expert in educational problem design.
You will be shown these components:
**Problem:** {problem}
**Fundamental Concepts:** {list of concepts}
**Difficulty Level:** {difficulty_level}
Your task is to reverse-engineer a clear thinking process that shows how a teacher might design this problem. This thinking process should:

- Show how combining the given foundational concepts naturally leads to a problem at the specified difficulty level.

- Include all key decisions and reasoning that shaped the problem design.

- **(IMPORTANT)** Be so precise and detailed that another teacher following these exact steps would recreate the identical problem.

- **(IMPORTANT)** Be so natural and logical that another teacher could derive the same thinking process using only the foundational concepts and difficulty level.

Present your answer after "Thinking Process:" with the complete step-by-step thinking process described above.

---

### B.3. Prompt Synthesis

**Code-start Initialization.** We initialized the framework using open-source problem collections, drawing 9,221 programming problems from Codeforces (Penedo et al., 2025) and 6,365 mathematics problems from AoPS[3]. For each problem, we annotated associated concepts and rationales by querying four high-capacity instruction-tuned models: `Qwen2.5-32B-Instruct`, `Qwen2.5-72B-Instruct`, `Llama-3.1-70B-Instruct` (Grattafiori et al., 2024) and `phi-4` (Abdin et al., 2024). The instructions used for concept extraction and rationale generation are provided in Appendix B.1 and Appendix B.2, respectively. These annotations provided the supervision to warm-start both the rationale generation model and the prompt generation model, both of which were initialized from `Qwen2.5-32B-Base`. Models were trained with a learning rate of $2 \times 10^{-5}$, a batch size of 16, and for two epochs.

---

[3]https://artofproblemsolving.com/

**EM Optimization.** In the EM refinement stage, the rationale generation model (E-step) was trained with a learning rate of $2 \times 10^{-6}$, a batch size of 16, and a sampling temperature of 1.0. In the M-step, rationales decoded deterministically from this model (temperature 0.0) were used to update the prompt generation model, which was trained with a learning rate of $2 \times 10^{-6}$ and a batch size of 16.

### B.4. Post-training with Synthesized Prompts

**Self-Play.** For the 4B models, we constructed a training corpus of 48,113 prompts, comprising 4,707 programming prompts and 43,406 mathematics prompts. Among the programming prompts, 1,872 were synthesized using PROMPTSCALE, while 2,835 were drawn from Codeforces (Penedo et al., 2025) and LiveCodeBench contest problems released prior to August 2024. For mathematics, 30,435 prompts were synthesized using PROMPTSCALE and 12,971 originate from DeepScaleR (Luo et al., 2025b). Each synthesized programming prompt was paired with 3–4 automatically generated unit tests using `Qwen3-32B`, while each synthesized mathematics prompt was paired with a final boxed answer obtained via majority voting over 8 generations from `Qwen3-30B-A3B-Thinking-2507`.

For the 30B-A3B models, we prepared a dataset of 11,209 prompts, including 3,072 programming prompts and 8,137 mathematics prompts. Within programming, 1,024 prompts were synthesized using PROMPTSCALE and 2,048 came from Codeforces and LiveCodeBench problems (prior to August 2024). In mathematics, 3,333 prompts were synthesized using PROMPTSCALE and 4,804 were sourced from DeepScaleR. Similarly, synthesized programming prompts were paired with 3–4 unit tests generated by `Qwen3-32B`, while mathematics prompts were paired with final boxed answers determined through majority voting over 8 generations from `GPT-OSS-120B (medium)`.

During self-play, we use a binary reward $u(x, y)$: $u(x, y) = 1$ if (i) the generated final answer exactly matches the boxed reference for mathematics, or (ii) the generated program passes all unit tests for programming; otherwise $u(x, y) = 0$. Self-play training was conducted with direct preference optimization (DPO) (Rafailov et al., 2023). For each prompt $x$, we sampled eight rollouts and evaluated them using the reward $u(x, y)$. Rollouts with $u(x, y) = 1$ were treated as positive samples, while those with $u(x, y) = 0$ were treated as negative samples. To ensure adequate task difficulty, we applied a filtering step: problems that the self-play models solved in at least half of eight independent attempts were excluded. Sampling temperatures were set to 1.25 for 4B models and 1.2 for 30B-A3B models respectively, as higher temperatures were found to substantially increase the proportion of invalid rollouts (e.g., formatting errors or corrupted outputs). Potential overlap with evaluation sets was mitigated by 13-gram matching to remove contaminated instances (Yang et al., 2024). All self-play training runs used a batch size of 16 and a learning rate of $1 \times 10^{-6}$.

**SFT.** For the SFT setting, we employed `GPT-OSS-120B (medium)` as the teacher model. All training prompts were synthesized using PROMPTSCALE, *without* incorporating external problem collections. We applied a light filtering step, discarding instances where the teacher failed to produce a valid final boxed answer (for mathematics) or an executable code snippet (for programming). This process produced a corpus of 4,766,890 prompts, consisting of 1,188,505 programming prompts and 3,578,385 mathematics prompts. Training was performed with a batch size of 1,024 and a learning rate of $8 \times 10^{-5}$.

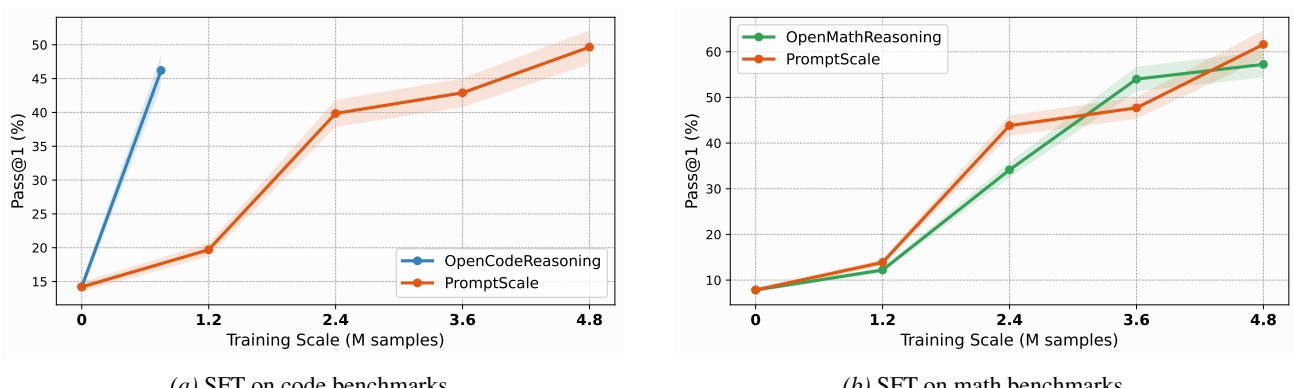

*(a)* SFT on code benchmarks.  *(b)* SFT on math benchmarks.

*Figure 3.* Learning curves of PROMPTSCALE compared with baseline methods.

## C. Additional Experimental Results and Analysis

### C.1. SFT Scaling Analysis for RQ2

Figure 3 (a) and Figure 3 (b) report the effect of increasing training data size from 0 to 4.8M examples. Here too, the difference between curation-based and rationale-driven synthesis is pronounced. While baselines exhibit sharp but plateauing gains, PROMPTSCALE achieves both higher peak performance and stronger data efficiency.

### C.2. Distributional Analysis for RQ3

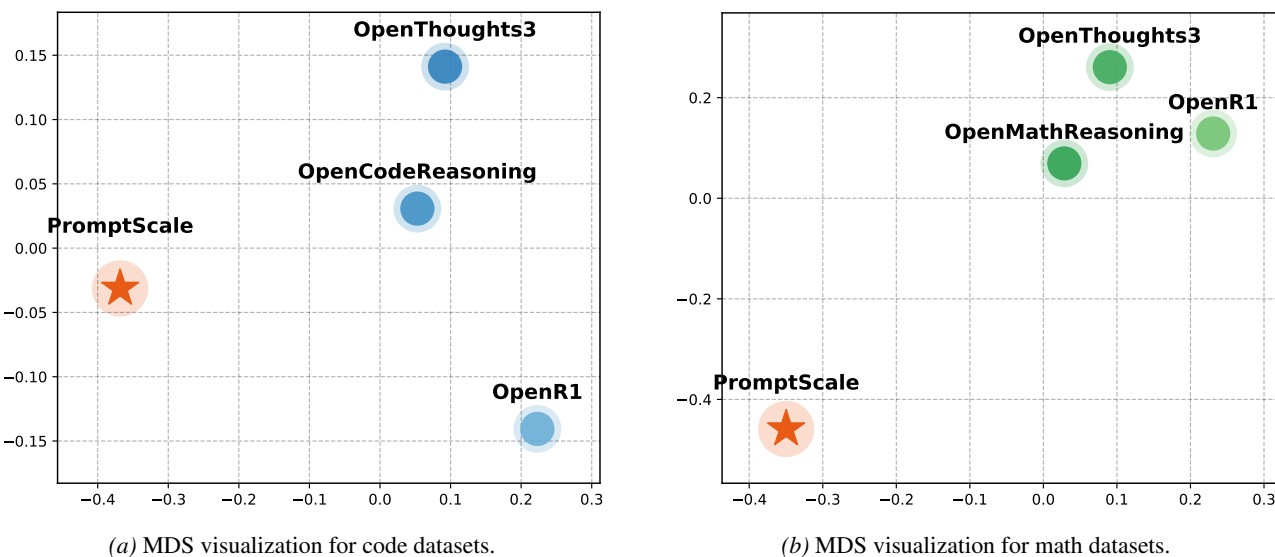

*(a)* MDS visualization for code datasets.    *(b)* MDS visualization for math datasets.

*Figure 4.* Multidimensional scaling (MDS) projections of dataset-level embeddings. Distances are based on cosine dissimilarity between average problem embeddings.

To investigated the distributional properties of our synthesized problems relative to existing open-source datasets, we computed embeddings for each problem using the `all-MiniLM-L6-v2` sentence transformer (Wang et al., 2020). For each dataset, we took the average embedding across all problems to obtain a dataset-level representation. Pairwise distances were then computed as $d(A, B) = 1 - \cos(A, B)$, where $\cos(\cdot, \cdot)$ denotes cosine similarity. We applied multidimensional scaling (MDS) to project these distances into two dimensions, yielding a visualization that captures relative relationships among datasets in a low-dimensional space.

The results, shown in Figure 4, reveal two notable findings: (1) Existing open-source datasets (e.g., OpenCodeReasoning, OpenMathReasoning, OpenThoughts3, and OpenR1) form tight clusters, reflecting their high similarity and shared reliance on either curated or slightly varied synthetic problems. (2) In contrast, our synthesized problems (PROMPTSCALE) occupy a distinct region in the embedding space, indicating substantial distributional differences from existing corpora. This separation suggests that our rationale-driven synthesis paradigm not only generates problems at scale, but also introduces novel linguistic and structural variations beyond the scope of prior datasets.

### C.3. Difficulty Analysis for RQ4

We evaluated dataset difficulty along two complementary axes. For each dataset, we uniformly sampled 1,000 prompts and (i) measure the *zero-shot* accuracy of `Qwen2.5-72B-Instruct` against reference final answers produced by `GPT-OSS-120B (high)`; and (ii) recorded the average number of reasoning tokens consumed by `GPT-OSS-120B (high)` when deriving those references. Both metrics are reported in Table 5.

From these results, we derive three key observations: (1) PROMPTSCALE produces the most challenging problems, reflected by both the lowest accuracy (18.5%) and the longest reasoning traces (37.4k tokens). This shows that our framework pushes problem difficulty substantially further than existing datasets. (2) Datasets such as OpenMathReasoning, OpenThoughts3, and PromptCoT apply explicit difficulty filtering, which raises the challenge level. Nevertheless, PROMPTSCALE surpasses

*Table 5.* Difficulty evaluation across datasets. Lower accuracy of `Qwen2.5-72B-Instruct` indicates higher difficulty, while longer reasoning traces from `GPT-OSS-120B (high)` suggest that problems require deeper reasoning. [†]: datasets that apply explicit difficulty filtering.

| Dataset | Qwen2.5-72B-Instruct Accuracy ($\downarrow$) | GPT-OSS-120B (high) Avg. Resp. Tokens ($\uparrow$) |
|---|---|---|
| **OpenMathReasoning**[†] | 28.9 | 18,436.9 |
| **OpenThoughts3**[†] | 21.3 | 30,053.7 |
| **OpenR1** | 32.3 | 7,128.3 |
| **PromptCoT**[†] | 24.7 | 29,425.5 |
| **PROMPTSCALE** | **18.5** | **37,373.3** |

them without such filtering, demonstrating that its generative process naturally produces harder problems.

## C.4. EM Optimization Analysis for RQ5

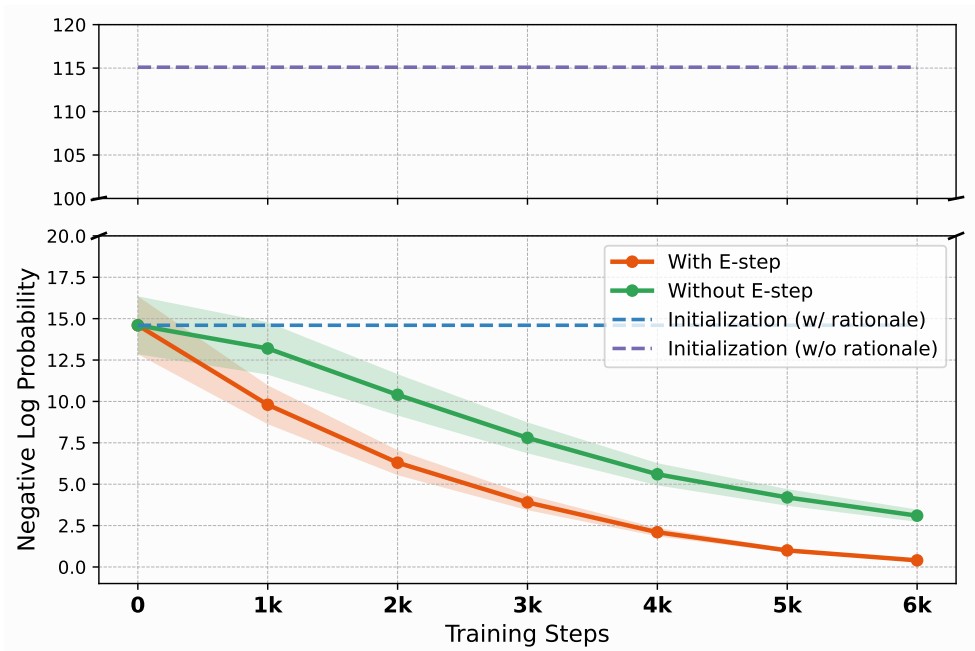

*Figure 5.* NLL trajectories during EM optimization. Curves compare training *with* and *without* the E-step, alongside initialization references with and without rationales. Confidence bands (shaded) are reported for the optimized variants.

We tracked the negative log-likelihood (NLL) of generating problems over training steps and compared four variants: (1) *With E-step*, the full EM procedure where rationales are iteratively refined by the approximate posterior; (2) *Without E-step*, where rationales are sampled once and fixed throughout optimization; (3) *Initialization (with rationale)*, a non-optimized reference that conditions on rationales; and (4) *Initialization (w/o rationale)*, the same starting point without rationale input. Confidence intervals are reported only for the two optimized variants to capture variability across repeated runs.

The results, shown in Figure 5, yield three key insights. (1) E-step enables sharper and deeper likelihood improvements. The *With E-step* curve consistently descends faster and achieves substantially lower terminal NLL compared to *Without E-step*, confirming the importance of posterior-guided rationale updates. (2) Rationales substantially reduce NLL even before optimization. The large gap between *Initialization (with rationale)* and *Initialization (w/o rationale)* highlights the structural advantage provided by rationales in connecting concepts to prompts. (3) Iterative refinement compounds the effect. Beyond early steps, the benefit of the E-step continues to widen, suggesting that iterative alignment between the posterior and the generative model progressively yields more informative rationales and sustained gains in likelihood.

*Table 6.* Evaluation results on six benchmarks under the SFT setting using models with **7B** parameters. **H** = Human-written, **S** = Synthetic, **Avg. Resp. Len.** = average response length, and **LCB** = LiveCodeBench. Bold values denote the best performance for each benchmark, while underlined values denote the second-best.

| Model | Prompt Source | Teacher Model | Dataset Size | Avg. Resp. Len. | AIME 24 | AIME 25 | HMMT Feb 25 | LCB v5 (2408–2502) | LCB v6 (2502–2505) | Codeforces |
|---|---|---|---|---|---|---|---|---|---|---|
| **Qwen2.5-7B-Instruct** | - | - | - | - | 12.8 | 8.0 | 2.7 | 14.7 | 13.7 | 706 |
| **OpenCodeReasoning** | H | GPT-OSS-120B (medium) | 0.75M | 7394.9 | 19.4 | 11.0 | 4.8 | 38.7 | 34.4 | 1574 |
| **OpenMathReasoning** | H | GPT-OSS-120B (medium) | 4.92M | 5788.9 | 65.4 | 55.0 | 40.6 | 7.2 | 6.9 | 437 |
| **OpenThoughts3** | H+S | GPT-OSS-120B (medium) | 1.20M | 8967.2 | 52.3 | 37.3 | 24.0 | 22.2 | 21.4 | 907 |
| **PROMPTSCALE** | S | GPT-OSS-120B (medium) | 4.77M | 7351.2 | **73.1** | **65.6** | **46.5** | **53.4** | **48.9** | **1815** |

## C.5. Controlled Comparison with Unified Teacher Model

To facilitate a controlled comparison between PROMPTSCALE and the baselines, we conducted an additional experiment where all baseline models were evaluated with the same teacher model, `GPT-OSS-120B (medium)`. This ensures that the differences in performance can be attributed to the respective methods rather than variations in the teacher model.

The results, as shown in Table 6, demonstrate the performance of various models under the SFT setting with 7B parameters, using the same teacher model across all baselines. In this setup, PROMPTSCALE consistently outperforms the other baselines across all six benchmarks, including AIME 24, AIME 25, HMMT Feb 25, LiveCodeBench v5/v6, and Codeforces. Notably, PROMPTSCALE achieves the highest scores in all benchmarks, with significant improvements over the best-performing baseline, particularly in AIME 24, AIME 25, and HMMT Feb 25.

## C.6. Case Study

**Foundational Concepts:**

1. Understanding the definition and properties of divisors of a number

2. Comprehending the concept of a function and its application to number theory, specifically the sum of divisors function $f(n)$

3. Ability to compute the sum of a sequence of numbers, including understanding the concept of cumulative sums as represented by $F(n)$

4. Knowledge of the parity (odd or even) of numbers and how it changes under addition, particularly in the context of sums of divisors

5. Application of modular arithmetic to determine conditions under which $F(n)$ is odd, especially modulo 2 operations

**Rationale:**
**Step 1: Identify the Foundational Concepts**
The problem uses divisors and the sum of divisors function (Concepts 1 and 2), cumulative sums (Concept 3), parity (Concept 4), and modular arithmetic (Concept 5).

**Step 2: Difficulty Level**
The problem is intended for the AIME, requiring multi-step reasoning in number theory.

**Step 3: Formulation**
Define $f(n)$ as the sum of divisors of $n$. Define $F(n) = \sum_{k=1}^{n} f(k)$. We want the count of $n \leq 2005$ for which $F(n)$ is odd.

**Step 4: Analysis of Parity**
The parity of $F(n)$ is determined by the parity of $\sum_{k=1}^{n} f(k)$. Thus, we must characterize when $f(k)$ is odd.

**Step 5: Characterizing Odd $f(k)$**
A number $k$ has an odd sum of divisors exactly when $k$ is a perfect square or twice a perfect square.

- This follows from the multiplicativity of $\sigma(n)$ and the fact that for odd primes $p$, $\sigma(p^e)$ is odd iff $e$ is even.

- Therefore, the odd part of $k$ must be a square, and the power of 2 can be 0 or 1.

**Step 6: Counting Relevant $k$**
Hence
$$F(n) \equiv \#\{k \leq n : k \text{ is a square}\} + \#\{k \leq n : k = 2m^2\} \pmod{2}.$$

Equivalently,
$$F(n) \equiv \lfloor \sqrt{n} \rfloor + \lfloor \sqrt{n/2} \rfloor \pmod{2}.$$

So $F(n)$ is odd exactly when $\lfloor \sqrt{n} \rfloor$ and $\lfloor \sqrt{n/2} \rfloor$ have opposite parity.

**Step 7: Solving the Problem**
For $n \leq 2005$:

- $\lfloor \sqrt{2005} \rfloor = 44$

- $\lfloor \sqrt{2005/2} \rfloor = 31$

Counting by intervals $n \in [t^2, (t+1)^2 - 1]$ for $t = 1, \ldots, 44$, and checking where $\lfloor \sqrt{n/2} \rfloor$ has opposite parity to $t$, gives the total number of integers $n \leq 2005$ with $F(n)$ odd:

$$\#\{n \leq 2005 : F(n) \text{ odd}\} = 1018.$$

**Final Problem:**
For any positive integer $n$, let $f(n)$ be the sum of the divisors of $n$, and let $F(n)$ be the sum of the numbers $f(k)$ for $k = 1$ to $n$. For example, $F(4) = f(1) + f(2) + f(3) + f(4) = 1 + 3 + 4 + 7 = 15$. Find the number of positive integers $n \leq 2005$ with $F(n)$ odd.

*Figure 6.* Case study of rationale and problem generation for a number theory AIME-level problem.

**Foundational Concepts:**

1. Understanding of the properties of a cube, including its edges, vertices, and faces

2. Knowledge of how a plane can intersect a three-dimensional solid, specifically a cube, and the types of polygons that can be formed

3. Ability to visualize and analyze geometric shapes in three dimensions, particularly the cross-sections of a cube

4. Understanding of the concept of perimeter and how to calculate it for different types of polygons, especially in the context of a plane intersecting a cube

5. Application of the Pythagorean theorem to calculate distances within the cube, which are essential for determining the perimeter of the intersection polygon

**Rationale:**

**Step 1: Core Concept and Difficulty Level**

The problem focuses on the intersection of a plane with a cube and the polygon formed by this intersection. The goal is to determine the maximum possible perimeter of such a polygon. This is appropriate for the AIME level, which requires multi-step reasoning in three dimensions.

**Step 2: Cube and Plane Setup**

We consider a cube of edge length 1. The intersecting plane is restricted from passing through any cube vertices, ensuring the cross-section is a polygon with between 3 and 6 sides.

**Step 3: Possible Intersections**

A plane intersecting a cube can produce polygons with 3, 4, 5, or 6 sides. To maximize perimeter, we must look for the configuration that yields the longest polygonal boundary.

**Step 4: Quadrilateral Case**

If the intersection is a quadrilateral, the largest perimeter arises when the cross-section is a square taken through the center of the cube. In this case each side has length 1, giving perimeter:
$$P = 4.$$

**Step 5: Hexagon Case**

When the plane is perpendicular to a space diagonal of the cube and passes through its center, the intersection is a regular hexagon. Each side of this hexagon has length $\frac{\sqrt{2}}{2}$, so the perimeter is:

$$P = 6 \cdot \frac{\sqrt{2}}{2} = 3\sqrt{2}.$$

**Step 6: Conclusion**

Comparing these cases, the hexagon provides the largest possible perimeter. Therefore:

$$\text{Maximum Perimeter} = 3\sqrt{2}.$$

**Final Problem:**

A plane intersects a cube with edge length 1. The plane does not pass through any of the cube's vertices. What is the maximum possible perimeter of the polygon formed by the intersection of the plane and the cube?

*Figure 7.* Case study of rationale and problem generation for a 3D geometry AIME-level problem.

**Foundational Concepts:**

1. Understanding of basic arithmetic operations, specifically addition, and the concept of carrying in multi-digit addition

2. Ability to manipulate and process arrays, including iterating through elements and performing operations on pairs of elements

3. Comprehension of nested loops and their use in computing pairwise operations over a range of elements in an array

4. Knowledge of modular arithmetic and digit manipulation to handle addition without carrying, focusing on individual digits

5. Understanding of summation notation and the ability to compute the sum of a series of values, particularly in the context of nested summations

**Rationale:**

**Step 1: Identify the Foundational Concepts**
The problem directly draws from addition without carrying (Concept 1), array iteration (Concept 2), nested loops (Concept 3), digit-wise modular arithmetic (Concept 4), and summation notation (Concept 5).

**Step 2: Define the Problem Context**
A student (Tot) adds numbers digit by digit without carrying. The task is to generalize this operation to arrays and compute the total over all pairs.

**Step 3: Problem Statement**
Given an array $a$ of length $n$, compute $\sum_{i=1}^{n} \sum_{j=i}^{n} f(a_i, a_j)$ where $f(x, y)$ is the digit-wise sum of $x$ and $y$ without carrying.

**Step 4: Input and Output**
The input consists of multiple queries. Each query specifies $n$ and then $n$ integers. For each query, output the required double summation.

**Step 5: Constraints**
The total $n$ across all queries is at most $10^5$, ensuring feasibility with efficient implementations.

**Step 6: Sample Input/Output**
Verified with examples:

- $f(53, 19) = 612$, matching digit-wise no-carry addition.

- Summations over arrays give large outputs consistent with constraints.

**Step 7: Difficulty Level**
At the Codeforces level, the challenge is balancing array operations and efficiency. A naive $O(n^2)$ summation may be too slow, so solutions must exploit structure.

**Step 8: Logical Flow**
The reasoning starts from digit-wise addition, extends to arrays, introduces nested summations, and ensures scalability under the constraints.

**Step 9: Final Problem Statement**
*Tot was doing homework in arithmetic. He was taught how to add two numbers, but not how to carry. For example, for $53 + 19$, he would do:*

$$\begin{array}{r} 53 \\ + \ 19 \\ \hline 612 \end{array}$$

*Tot was given an array $a$ of length $n$. He was asked to find $\sum_{i=1}^{n} \sum_{j=i}^{n} f(a_i, a_j)$, where $f(x, y)$ is the number obtained by adding $x$ and $y$ without carrying. For example, $f(53, 19) = 612$.*

**Input:**
The first line contains one integer $t$ ($1 \le t \le 1000$) — the number of queries. Each query begins with an integer $n$ ($1 \le n \le 10^5$). The next line contains $n$ integers $a_1, \ldots, a_n$ ($1 \le a_i \le 10^9$). The total $n$ over all queries does not exceed $10^5$.

**Output:**
For each query, print the required sum.

*Figure 8.* Case study of rationale and problem generation for a Codeforces-level programming problem.

