# OpenReview forum: "Scaling Prompt Synthesis for Large Language Model Reasoning"
_ICML.cc/2026/Conference — ICML 2026 regular_

### Official Review · Reviewer_atAB · 2026-03-12

**Soundness:** 3
**Presentation:** 3
**Significance:** 3
**Originality:** 3
**Overall Recommendation:** 4
**Confidence:** 3

**Summary:**

This paper studies the difficulty of scaling up high-quality training questions during the post-training stage of reasoning LLMs and proposes PROMPTSCALE. Methodologically, the paper models prompt synthesis as a latent-variable generation process involving concepts, rationales, and prompts, and jointly optimizes rationale generation and prompt generation through an approximate EM-style iterative framework. Based on the synthesized prompts, the authors further investigate two post-training settings: one is self-play for stronger models, where verifiable feedback is used for self-improvement; the other is SFT for weaker models, where a teacher model provides reasoning traces for distillation. The experiments cover multiple benchmarks in mathematics and coding, and the paper supports its claims with main results, ablations, scaling analysis, and analyses of difficulty and distribution. Overall, the paper aims to show that high-quality prompt synthesis can itself serve as an important scaling path for improving reasoning capability.

**Compliance With Llm Reviewing Policy:**

Affirmed.

**Key Questions For Authors:**

- Could the authors add a stricter matched-control comparison in the self-play setting, where the mixed data, verification signal, and training budget are kept as fixed as possible, and only the prompt synthesis module is changed? This would significantly strengthen the assessment of the core methodological contribution.

- For the SFT setting, could the authors further provide a more budget-normalized comparison, for example by comparing different prompt sources under as similar a data scale or token budget as possible?

- Could the authors more clearly explain the relationship between the current implementation and the EM/ELBO framing in the main text, especially the correspondence among rationale sampling, rationale selection, and the subsequent training stages?

- Could the authors provide more transparent statistics about the data-processing pipeline, such as contamination removal, difficulty filtering, and the potential noise introduced by automatically generated boxed answers and unit tests?

- For benchmarks with relatively small sample sizes, such as AIME and HMMT, could the authors provide more direct evidence of statistical robustness, such as bootstrap intervals or variance across multiple training runs?

**Limitations:**

No.

The paper does not contain a dedicated discussion of limitations. It would be helpful to more explicitly discuss the method’s boundary conditions, the role of mixed training data in self-play, the noise introduced by automatically generated boxed answers and unit tests, and the additional compute cost of iterative prompt synthesis.

**Strengths And Weaknesses:**

**Soundness**

**Strengths**

The problem setting is important, and the methodological perspective is fairly well developed rather than being a simple data-engineering exercise. The experiments cover both self-play and SFT, spanning the two main reasoning domains of mathematics and coding. Beyond the main results, the paper also provides ablations, scaling analysis, and a unified-teacher comparison in the appendix. Taken together, these results largely support the core claim that PROMPTSCALE-generated data has substantial training value.

**Weaknesses**

- The attribution of the core gains is still not fully clean. Especially in the self-play setting, the current results more safely support the claim that the overall training recipe is effective, rather than strictly isolating and validating the PROMPTSCALE mechanism itself.

- In the main SFT table, the teacher model, data scale, and response length are not strictly matched. Although the appendix adds a unified-teacher comparison, comparability at the level of the main table remains somewhat limited.

- There is still some gap between the EM and ELBO framing in the paper and the actual approximate implementation. A more direct explanation of this relationship would make the technical claim more convincing.

**Presentation**

**Strengths**

The paper is clearly organized. Figure 1 is helpful for understanding the method, and the experiments are presented in a fairly standard and easy-to-follow way. Readers can quickly grasp the problem, the method, and the main chain of evidence.

**Weaknesses**

Some key information that directly affects the interpretation of the results, such as the mixed data sources in self-play and the imperfect alignment of comparisons in the main SFT table, would be better stated more explicitly in the main paper rather than relying primarily on the appendix.

**Significance**

**Strengths**

This work addresses a very real bottleneck in reasoning LLMs, namely the difficulty of continuously scaling high-quality training questions. If this direction continues to hold up under stricter controls, it could have practical impact on reasoning data synthesis and post-training practice.

**Weaknesses**

The strongest evidence in the current paper is still concentrated on competition-style mathematics and coding tasks. Accordingly, I am more inclined to view the contribution as a competitive advance in reasoning data synthesis, rather than as a fully established new paradigm for reasoning scaling in a broader sense.

**Originality**

**Strengths**

The main novelty lies in organizing concepts, rationales, and prompts into an iteratively optimized framework and applying it to both self-play and SFT post-training settings. Treating the question-generation process itself as a learnable and optimizable object is an interesting overall perspective.

**Weaknesses**

At present, the novelty is more evident in the systematic framework integration and its empirical effectiveness. At the same time, the current experiments are still insufficient to fully disentangle whether the gains mainly come from the EM-style iterative refinement itself or from the combined effect of the overall training pipeline.

---

### Official Review · Reviewer_UMn2 · 2026-03-12

**Soundness:** 3
**Presentation:** 3
**Significance:** 3
**Originality:** 3
**Overall Recommendation:** 5
**Confidence:** 4

**Summary:**

The paper introduces an EM-based method to generate synthetic problems, which can be used to do post-training via self-play (RLVF) or SFT.   It is able to deliver synthetic problems fundamentally different from existing open-source datasets, thus helping post-training.

**Compliance With Llm Reviewing Policy:**

Affirmed.

**Key Questions For Authors:**

See Weakness 1

**Limitations:**

The authors have not  discussed the limitation of the proposed work, e.g, in which situation, it might not work.  It will be good to include one.

**Strengths And Weaknesses:**

Strength 1: The paper is well written and easy to follow.   It clearly identified the potential improvement that can be done over the previous method like PromptCoT that relies on human crafted instruction.

Strength 2: The experimental results demonstrate that those synthetically generated problems can significantly improve the performance on a few set of reasoning tasks.

Strength 3: The ablation study answered a few key questions like contribution of each component, the impact of increasing data size, etc.

Weakness 1: The experiments are all done on problem-centric benchmarks.  Since the title is Prompt Synthesis, do the authors have user cases other than math and coding problems?

---

### Official Review · Reviewer_cy9B · 2026-03-12

**Soundness:** 3
**Presentation:** 3
**Significance:** 3
**Originality:** 3
**Overall Recommendation:** 4
**Confidence:** 4

**Summary:**

The paper proposes PromptScale, a framework for synthesizing challenging math and programming problems by jointly training a rationale generation model and a prompt generation model through EM loop. The synthesized prompts support two post-training regimes: self-plan and SFT, both showing very promising results.

**Compliance With Llm Reviewing Policy:**

Affirmed.

**Key Questions For Authors:**

Q1: How many unique concepts are extracted? What's the distribution? Any dominating concepts? Is there a long tail of concepts?
Q2. How many EM iterations are needed to convergence?

**Strengths And Weaknesses:**

The strengths of the paper:
S1. The paper present a principled framework with thorough design. The EM loop is well-motivated, using rationales as the learnable latent variables that mediate between concepts and concrete problems. The presented solution is more principled than the heuristic prompt engineering in prior work.
S2. The experiment results are very promising. The trained models achieved SOTA in math domain. And the SFT results show a 7B model trained solely on synthetic prompts surpassing models trained on human curated datasets. The experiments also validate both self-play and SFT solutions, broadening the impact.
S3. The paper conducted thorough ablations and analysis, showing both cold-start and EM stages are necessary.

The weakness of the paper
W1. Only cover competition math and programming. These are domains with clean verifiable rewards. The paper does not discuss with domains where verification is harder, like logical reasoning, commonsense reasoning. Therefore the claim of "domain-agnostic" is not convincing.
W2. The paper showed Promptscale generates harder problems, but did not study the validity of the questions. How much of them are solvable, well-posed? Do we have malformed questions? Show the results on failure cases would be helpful to get the full picture of the solution.

---

### Official Review · Reviewer_aozs · 2026-03-14

**Soundness:** 3
**Presentation:** 3
**Significance:** 4
**Originality:** 3
**Overall Recommendation:** 4
**Confidence:** 4

**Summary:**

The paper proposes PROMPTSCALE a framework for synthesizing reasoning prompts for large language models by introducing rationales as latent variables and training prompt and rationale generators through an EM-style optimization loop. Experiments on math and programming benchmarks show that models trained with these synthesized prompts improve reasoning performance in both self-play and supervised fine-tuning settings.

**Compliance With Llm Reviewing Policy:**

Affirmed.

**Key Questions For Authors:**

Please refer to the weaknesses.

**Limitations:**

yes

**Strengths And Weaknesses:**

**Strengths**
- The paper tackles an important problem of scalable generation of high-quality reasoning tasks for LLM training.
- The work introduces an interesting prompt synthesis method as a promising direction for scaling reasoning capabilities beyond traditional human-curated datasets.

**Weaknesses**

*[W1]:*  The motivation for using EM over simpler alternatives is not fully convincing (besides the latent rational formulation). In particular the paper does not include comparisons/ablations with alternative training strategies such as direct prompt generation, iterative filtering etc. making it difficult to assess whether the EM formulation provides unique benefits (especially given the additional complexity and potential computational overhead introduced by the iterative EM optimization). How do you think EM brings unique benefits compared to other approaches?

*[W2]:* The qualitative analysis of generated prompts is limited. Providing concrete examples and comparisons with prompts from baseline datasets would help clarify the types of reasoning structures learned and better illustrate the benefits of the proposed approach.

*[W3]:* While the proposed framework is interesting, the overall pipeline may be computationally expensive to replicate (annotations generated by several large instruction-tuned teacher models) without access to such models and it is unclear whether the authors plan to release the synthesized dataset or the fine-tuned models.

*[W4]:* The paper provides limited discussion of the training dynamics of the ΕΜ process such as convergence behavior, sensitivity to initialization or instabilities during iterative rationale refinement.

---

### Decision · Program_Chairs · 2026-04-30

**Decision:**

Accept (regular)

**Comment:**

I thank the authors for their interesting work!  This paper develops a data synthesis strategy by training prompt and rationale generators via EM loop, resulting in diverse and hard problems that improve performance on math and code benchmarks.  Reviewers agreed that this paper approaches a timely problem, is well motivated and principled, and contains strong empirical results and solid ablations.  At the same time, reviewers pointed out the following potential weaknesses:

1. Only focuses on competition math and coding
2. Not clear where exactly the gains come from
3. Unclear the quality of generated prompts
4. Concerns about fair comparisons
5. Training stability is unclear
6. Small benchmarks without sufficient statistical analysis

The authors did miss the rebuttal deadline, but they did address reviewer feedback via AC note, and I am taking this note into strong consideration.  The authors presented additional comparisons and also did a manual review of prompts, finding that more than 80% are clean and solvable.  The authors further noted stable convergence of the EM algorithm.  Given the already positive feedback from reviewers, I am inclined to accept this paper.